# Rational design of a robust aluminum metal-organic framework for multi-purpose water-sorption-driven heat allocations

Kyung Ho Cho[1,8], D. Damasceno Borges [2,3,8], U-Hwang Lee [1,8], Ji Sun Lee[1], Ji Woong Yoon[1], Sung June Cho [4], Jaedeuk Park[1], Walter Lombardo[5], Dohyun Moon [6], Alessio Sapienza [5], Guillaume Maurin[2✉] & Jong-San Chang[1,7✉]

Adsorption-driven heat transfer technology using water as working fluid is a promising eco-friendly strategy to address the exponential increase of global energy demands for cooling and heating purposes. Here we present the water sorption properties of a porous aluminum carboxylate metal-organic framework, $[Al(OH)(C_6H_3NO_4)]\cdot nH_2O$, KMF-1, discovered by a joint computational predictive and experimental approaches, which exhibits step-like sorption isotherms, record volumetric working capacity (0.36 mL mL$^{-1}$) and specific energy capacity (263 kWh m$^{-3}$) under cooling working conditions, very high coefficient of performances of 0.75 (cooling) and 1.74 (heating) together with low driving temperature below 70 °C which allows the exploitation of solar heat, high cycling stability and remarkable heat storage capacity (348 kWh m$^{-3}$). This level of performances makes this porous material as a unique and ideal multi-purpose water adsorbent to tackle the challenges of thermal energy storage and its further efficient exploitation for both cooling and heating applications.

[1] Research Group for Nanocatalyst and Center for Convergent Chemical Process (CCP), Korea Research Institute of Chemical Technology (KRICT), Gageong-Ro 141, Yuseong, Daejeon 34114, South Korea. [2] Institut Charles Gerhardt, Montpellier UMR 5253 CNRS ENSCM UM, Université Montpellier, 34095, Montpellier CEDEX 05, France. [3] Instituto de Física, Universidade Federal de Uberlândia, Uberlândia-MG 38408-100, Uberlândia, Brazil. [4] Clean Energy Technology Laboratory and Department of Chemical Engineering, Chonnam National University, Gwangju 61186, Republic of Korea. [5] Consiglio Nazionale delle Ricerche (CNR), Istituto di Tecnologie Avanzate per l'Energia "Nicola Giordano" (ITAE), S. Lucia Sopra Contesse 5, 98126 Messina, Italy. [6] Beamline Department Pohang Accelerator Laboratory (PAL), Pohang, Gyeongbuk 37673, South Korea. [7] Department of Chemistry, Sungkyunkwan University, Suwon 440-476, South Korea. [8] These authors contributed equally: Kyung Ho Cho, D. Damasceno Borges, U-Hwang Lee. ✉email: guillaume.maurin1@umontpellier.fr; jschang@krict.re.kr

Global energy consumption of residential areas for heating and cooling has increased steadily over the last decades and is expected to rapidly grow in the near future[1]. The current supply system relies largely on non-sustainable energy resources in sharp discordance with recent international agreements that call for energy reduction generated by fossil fuels[2–4]. Therefore, the development of disruptive materials and systems implying clean and renewable energy resources is of utmost importance to address major societal concerns, such as global warming and drinking water supply[4–7]. Among them, adsorption-based heat transfer (AHT) devices like adsorption-driven cooling/chiller (AC) and adsorption-driven heat pump (AHP) systems actually emerge as cutting-edge alternatives[4,8]. They target a drastic reduction of energy consumption for cooling and heating owing to the potential use of natural solar or waste heat from industrial plants[4,9–14]. Moreover, water is an environmental-friendly natural refrigerant, non-flammable, abundant, cost-free, and has substantial latent heat of evaporation ($40.7 \, kJ \, mol^{-1}$)[11,14]. The use of water as working fluid in AHT devices has thus attracted tremendous interest in recent years[4,9–11,14–24]. AHT system typically operates under a full cycle of water adsorption/desorption (Supplementary Fig. 1a)[11,14]. During the adsorption step, heat is taken up from the evaporator ($Q_{ev}$), leading to a decrease in temperature enabling AC use, whereas when adsorption is exothermic, heat ($Q_{ads}$) is released to cooling water of the adsorbent bed. When the adsorbent becomes water-saturated, regeneration is achieved by the input of useful heat ($Q_{des}$) and the heat is released to cooling water of condenser ($Q_{con}$). AHP can thus operate with the released heats of adsorption ($Q_{ads}$) and condensation ($Q_{con}$). A detailed thermodynamic T–P diagram of such cycles is provided in Supplementary Fig. 1b[4]. Suitable temperature boundaries for evaporation ($T_{ev}$), adsorption ($T_{ads}$), condensation ($T_{con}$), and desorption ($T_{des}$) need to be identified to use the full capacity of the water adsorbents for AHT uses.

Conventional water adsorbents have clear drawbacks for AHT applications such as either low hydrophilicity (e.g., silica gel) or too high hydrophilicity (e.g., aluminosilicate zeolites) leading to water adsorption at rather high relative pressure ($P/P° > 0.3$) or the need for high regeneration temperature ($T > 150 \, °C$)[11,14,15,25]. As an alternative, SAPO-34 is a commercially usable adsorbent for AC devices due to its high water uptake at low pressure ($P/P° < 0.1$), combined with a very high durability[26]. However, the regeneration of this material at ~90 °C is not effectively compatible with the desired use of low-temperature heat source typically below 80 °C (solar energy) or even 70 °C (industrial waste heat) and its low working capacity ($0.20 \, g_{H2O}/g$) is a drastic limitation[11,15,18]. This calls for the development of advanced water adsorbents able to meet all the following five criteria under the specific AC/AHP operating conditions: (i) high working capacity on both volumetric and gravimetric basis; (ii) high coefficient of performance (COP), which is an indicator of an efficient thermal energy transfer; (iii) rapid water adsorption and desorption; (iv) regeneration at low temperature ($T < 70–80 \, °C$); and (v) good chemical and mechanical robustness, particularly being resistant to degradation in hot water vapor as a prerequisite for a long-term use[11,27]. The small-pore aluminophosphate AlPO-LTA (or AlPO-42) has recently been reported as a promising candidate to fulfill most of the requirements; however, its very limited stability is still a severe limitation for its potential use[16]. In this context, the metal–organic frameworks (MOFs) have been envisaged as next-generation water adsorbent, probably the most promising prospects to promote this family of materials at the market[4,11,14,15,17,18,20–23,28–34]. Since the pioneer works reported on the energy-efficient mesoporous metal(III) poly-carboxylates MIL-100 and MIL-101[35–38], Al-MOFs have received much attention owing to their attractive performances in terms of high working capacity, low regeneration temperature below 80 °C, and durability in multicycle adsorption–desorption experiments combined with the use of cheap (e.g., isophthalic acid for CAU-10)[17] or ecofriendly biomass-derived organic linkers (e.g., fumaric acid for MIL-53-FA and 2,5-furandicarboxylic acid for MIL-160)[15,18,30], low toxicity and abundance of Al precursor. Typically, MIL-53-FA, CAU-10, and CAU-23 have shown favorable features for AC devices owing to their low regeneration temperatures ($T = 60–70 \, °C$); however, their water working capacities are still too low for a desired cooling temperature below 10 °C[15,17,30,31]. Further, all these existing Al-MOFs show relatively modest water uptakes at low relative pressure $P/P° < 0.18$, which makes them inefficient in AHP devices[17,30,31]. Another Al-MOF, MOF-303 was reported as an efficient water harvesting material showing huge water uptakes at low relative pressure $P/P° < 0.2$; however, its potential application to AHT devices has never been evoked to date[39,40]. Besides Al-MOFs, Co-CUK-1 was demonstrated recently as the most promising MOF candidate for AHT devices[11], however, its working capacity still needs to be improved and the use of Co is not environmentally benign. Furthermore, its synthesis conditions and product yields are still far to be optimal (Supplementary Table 1). Therefore, the discovery of more efficient MOF water adsorbents to target multipurpose materials with remarkable performances in both AC and AHP technology is still a challenge.

As a generalization of our concept, first developed to discover MIL-160, the 2,5-furanedicarboxylic acid (FDCA) analog of CAU-10 type structure[15,17], here we report the rational design of a channel-like Al-MOF incorporating 2,5-pyrroledicarboxylic acid (PyDC), denoted KMF-1 (KMF = **K**RICT-CNRS-**M**ontpellier **F**ramework) prepared by a green and high-yield (>93%) synthesis route. This potentially easy scalable, i.e., a good space-time to yield (STY) of 68 kg m$^{-3}$ day$^{-1}$, and highly stable material exhibits a large water uptake of 0.381 g$_{H2O}$/g$_{MOF}$ at 30 °C in a low relative pressure range ($P/P° = 0.13–0.15$), facile desorption at low temperature ($T < 70 \, °C$) and excellent stability over at least 50 adsorption/desorption cycles. More importantly, KMF-1 shows an exceptional level of performance for both AC and AHP systems with a coefficient of performance for cooling (COP$_C$) and for heating (COP$_H$), specific energy capacity (Q$_{ev}$) and energy-storage capacity (Q$_{stored}$) that outperform not only all the best MOFs reported so far but also the commercial state-of-the-art water adsorbents for each AHT application. This remarkable capability makes KMF-1 as a unique multipurpose water adsorbent that encompasses the optimal features to cover the full spectrum of water-adsorption-driven applications.

## Results

**Computational design toward green synthesis**. We deliberately built in silico an isostructural analog of CAU-10 with the objective to achieve a steep water-adsorption isotherm at a relative pressure intermediate between $P/P° = 0.05$ and $P/P° = 0.17$ exhibited by the furane-dicarboxylate-based CAU-10 derivative (MIL-160)[15] and the benzene-dicarboxylate CAU-10[15], respectively, while maintaining a water saturation capacity as high as for MIL-160. This optimal target range for $P/P°$ was identified from our previous findings on the best MOF water adsorbents so far, i.e., Co-CUK-1[11] ($P/P° = 0.12$) of similar pore size (~6 Å) than the CAU-10 platform. Indeed, we selected in purpose the 2,5-pyrroledicarboxylate (PyDC) linker instead of the pristine benzene-dicarboxylate linker with the main objective, besides achieving a higher water uptake with the use of a five-membered heterocycle, to fine-tune the hydrophilicity of the pore environment by the introduction of polar −NH functional groups

capable of hydrogen bonding interactions expected to be of weaker strength than those established by the bridging μ-OH groups of the inorganic node that rendered MIL-160 too hydrophilic.

This 1D-channel-like architecture (Supplementary Fig. 2 and Table 2) of relatively large pore volume (theoretically 0.56 cm³ g⁻¹) with polar N-functions potentially accessible to water was further predicted by Grand Canonical Monte Carlo (GCMC) simulations to be an optimal adsorbent achieving steep water adsorption at a low relative pressure ($P/P°{\sim}0.12$) and a large water uptake over 0.48 $g_{H2O}$/$g_{MOF}$ at $P/P° > 0.5$ (Supplementary Fig. 3). These computational findings encouraged us to devise a strategy to prepare this heterocyclic Al dicarboxylate MOF. The green and high-yield (93%) synthesis of the compound $[Al(OH)(C_6H_3NO_4)]$ $nH_2O$ ($n = 0$ and 4.5 for the anhydrous and hydrated forms, respectively) (see elemental analysis on Supplementary Table 3), denoted KFM-1, was achieved hydrothermally with the use of green chemicals, aqueous solvent under atmospheric reflux condition, and mild temperature of 120 °C. This protocol led to an easy multi-gram-scale production of highly crystalline KMF-1 with particle sizes ranging between 0.4 and 1 μm as displayed by scanning electron microscopy (SEM) images (Supplementary Fig. 4). A relatively good STY over 68 kg m⁻³ day⁻¹ was obtained after simple purification with water and ethanol, which is much higher than the value previously reported for the best water adsorbent Co-CUK-1 so far (24 kg m⁻³ day⁻¹) (Supplementary Table 1). Importantly, this material was shown to be potentially easily scalable since there is no significant difference between resulting products of KMF-1 after synthesis in 50-mL and 500-mL reaction scales.

The crystal structures of KMF-1 for the hydrated (Fig. 1) and anhydrous (Supplementary Fig. 5) samples were solved using a powder charge flip method[41] and then refined with Jana2006 package[42] over the corresponding synchrotron powder diffraction data. The hydrated form was indexed in the non-centrosymmetric tetragonal space group ($I4_1md$) with unit cell parameters of $a = 21.1772(2)$ Å and $c = 10.70115(17)$ Å (Supplementary Table 4) in excellent agreement with those of the in silico constructed structure (Supplementary Fig. 2). The PyDC linker was modeled using the Translation–Libration–Screw (TLS) method[43] in order to minimize the number of refined parameters. The water molecules were located through Fourier difference allowing the distinction of six independent water positions. The oxygen–oxygen distances between the guest water are in the range of 2.56–2.97 Å, indicating the formation of a confined water hydrogen-bonded network (Supplementary Fig. 6). In terms of host/guest interactions, one water oxygen is located at 2.73(4) Å from the oxygen of the μ-OH groups of the inorganic building unit, while another water oxygen is separated to an oxygen of the carboxylate from the pyrrole linker by a distance of 3.02(3) Å. The resolved anhydrous structure of centrosymmetric tetragonal symmetry ($I4_1/amd$) shows very similar cell parameters ($a = 21.225(2)$ Å and $c = 10.6424(16)$ Å) to those of the hydrated version (Supplementary Table 4).

KMF-1 is made of inorganic cis-corner-sharing octahedra Al chains coordinated to carboxylate groups of the PyDC linker. All polyhedra are linked to oxygen atoms from four PyDC linkers and two hydroxyl groups, which are in cis-position and bridge the Al centers to create the chains. The PyDC linkers are bounded to four octahedra from two chains together. This leads to a 3D framework determining square-shaped sinusoidal 1D channels of about 5.7 Å in diameter (Fig. 1b). KMF-1 shows a permanent porosity with a BET area and a pore volume of 1130 m² g⁻¹ and 0.473 cm³ g⁻¹ estimated from the nitrogen-adsorption isotherm at −196 °C (Supplementary Fig. 7) consistent with the theoretical values of 1290 m² g⁻¹ and 0.56 cm³ g⁻¹, respectively, calculated from the predicted crystal structure. Further thermogravimetric

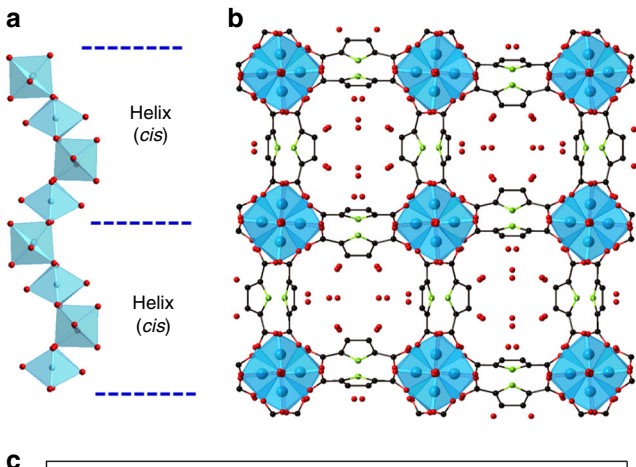

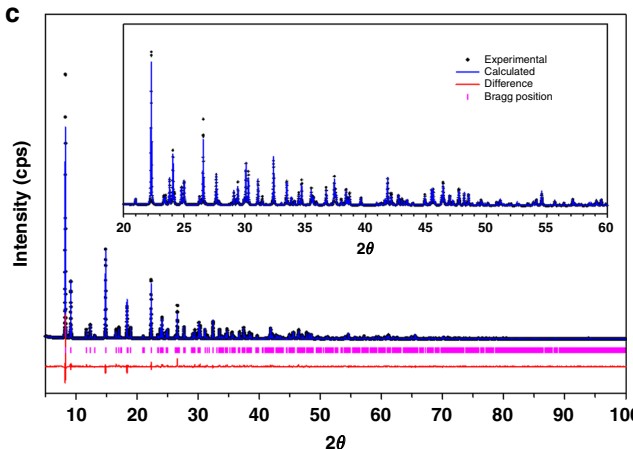

**Fig. 1 Crystal structure determination of KMF-1. a** Helical chains cis-connected Al octahedra by the corner and coordinated by four carboxylates from the linker along the a axis. **b** Full crystal structure of KMF-1 in its hydrated form (water positions included) with a square-shaped channel aligned along the c axis. **c** Rietveld refinement of the KMF-1 hydrated structure model. The experimental data are shown in black, simulation in blue, and difference in red below. Allowed Bragg reflections are indicated as magenta ticks below. Color scheme: Al, blue; C, black; O, red; N, green (hydrogen atoms omitted for clarity).

(TG) analysis and high-temperature powder X-ray diffraction (HT-PXRD) experiments revealed that KMF-1 is thermally stable up to 300 °C (Supplementary Fig. 8, 9). Further, besides a very good hydrothermal stability (Supplementary Fig. 10), KMF-1 maintains the integrity of its crystal structure in boiling water and in a wide range of acidic and basic aqueous solutions (1 < pH < 12) at room temperature for 1 day as evidenced by identical PXRD patterns and nitrogen-adsorption isotherms under each tested condition (Supplementary Fig. 11 and Table 5). This combined hydrothermally and chemically remarkable stability makes KMF-1 suitable for AHT applications.

**Water sorption.** The water-adsorption behavior of KMF-1 was first explored at three different temperatures (20–40 °C) (Fig. 2a) revealing an S-shaped adsorption isotherm with a full reversible desorption branch. Typically, steep water uptake of 0.393 $g_{H2O}$ $g_{MOF}$⁻¹ at 30 °C was obtained in the range of $P/P° = 0.13$–0.20, which outperforms all benchmark materials below $P/P° = 0.2$ (Fig. 2b), including SAPO-34 (0.282 $g_{H2O}$ $g_{MOF}$⁻¹ at $P/P° = 0.2$)[4], MOF-303 (0.373 $g_{H2O}$ $g_{MOF}$⁻¹ at $P/P° = 0.2$)[40], CAU-23 (0.334 $g_{H2O}$ $g_{MOF}$⁻¹ at $P/P° = 0.3$)[31], CAU-10

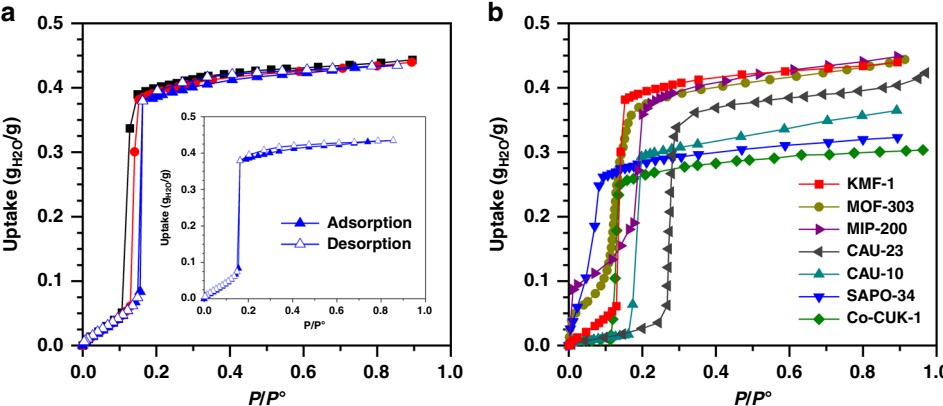

**Fig. 2 Water-sorption isotherms for KMF-1 and benchmark adsorbents. a** Experimental water-adsorption isotherms for KMF-1 collected at three different temperatures: 20 °C (■), 30 °C (●), and 40 °C (▲). Insert shows the adsorption/desorption isotherms at 40 °C. **b** Comparison of water-adsorption isotherms for KMF-1 (■) and benchmark adsorbents: MOF-303 (●)[40], CAU-10 (▲)[4], SAPO-34 (▼)[4], MIP-200 (▶)[14], Co-CUK-1(◆)[11] at 30 °C, and CAU-23 (◀)[31] at 25 °C.

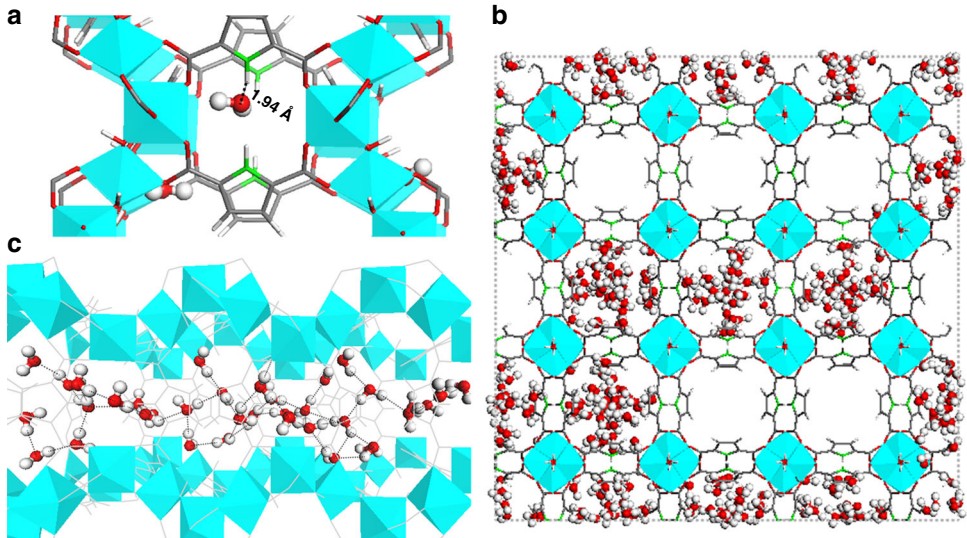

**Fig. 3 Microscopic water adsorption mechanism in KMF-1.** Preferential arrangements of the water molecules in the channel of KFM-1 at different $P/P^0$ (**a**) 0.10, (**b**) 0.14, and (**c**) 1.00. **a** Interaction between the very first adsorbed water molecules and the NH function of the linker. **b** Gradual filling of the porosity with water molecules adsorbed channel by channel. **c** Formation of a strong H-bond network when the pore is saturated, the black dash lines indicating the H bonds.

(0.293 $g_{H2O}$ $g_{MOF}^{-1}$ at $P/P° = 0.2$)[4], as well as the Zr-based MIP-200 (0.360 $g_{H2O}$ $g_{MOF}^{-1}$ at $P/P° = 0.2$)[14] and Co-CUK-1 (0.263 $g_{H2O}$ $g_{MOF}^{-1}$ at $P/P° = 0.2$)[11] all these corresponding data being extracted from the literature[4,11,14,31,40].

This experimental profile is in excellent agreement with the GCMC-simulated adsorption isotherm (Supplementary Fig. 3), the deviation between theoretical and experimental data at very low pressure being mostly associated with water adsorption at residual defect sites and/or external surface of the MOF. A careful analysis of the adsorption mechanism (Supplementary Fig. 12) reveals that the water adsorption occurs initially in the vicinity of the linker (Fig. 3a). Analysis of the radial distribution function (RDF) calculated for the corresponding intermolecular pairs emphasizes that the oxygen atom of $H_2O$ (Ow) preferentially interacts with the H atom bound to the –N atom (Hn) of the PyDC linker (Supplementary Fig. 13) with a mean separating distance of 2.02 Å. The population of water molecules further continues to rise until corresponding to a monotonic increment of the water molecules in the region nearby the linker. This is followed by a sudden increase

in the water content above $P/P° = 0.14$ and the water molecules tend to fill up KFM-1 channel by channel (Fig. 3b), forming H bonds between each other. At saturation (above $P/P° = 0.22$), water molecules also interact with the μ-OH sites of the Al octahedral chains as shown by the RDF plot for the corresponding pair H (w)–O(μ-OH) associated with a short mean separating distance of 1.7 Å (see Supplementary Fig. 14) consistent with the experimental findings described above for the resolved hydrated structure. This leads to the formation of a strong hydrogen-bond network (Fig. 3c) with a total number of hydrogen bond per water molecule of 2.7 (see Supplementary Fig. 15) similar to the value previously reported for other MOF/water systems in the same conditions[11,31,44].

A moderate isosteric heat of water adsorption ($-\Delta_{ads}H$) ranging from 52 to 57 kJ mol$^{-1}$ (Supplementary Fig. 16) was derived by applying both the Clausius–Clapeyron equation and the virial equation[45] to the water adsorption isotherms collected at three different temperatures (Fig. 2a). This value is consistent with the adsorption enthalpy predicted by GCMC simulations at 0.1 $g_{H2O}$ $g_{MOF}^{-1}$ ($-54$ kJ mol$^{-1}$), suggesting a facile regeneration of KMF-1.

Cycling tests using regeneration temperature ranging from 70 °C to 100 °C further revealed that the material maintains its remarkable water-adsorption performance over 50 adsorption/desorption cycles (Supplementary Fig. 17). This observation encouraged us to examine the performances of KMF-1 in different scenarios of AC and AHP applications.

**Assessment of the performances of KFM-1 under operating AC and AHP conditions**. Material-related COPs are standard metrics to assess the efficiency of water adsorbents in AC and AHP applications. They are evaluated by thermodynamic models applied at different boundary temperature conditions for water evaporation ($T_{ev}$), condensation ($T_{con}$), adsorption ($T_{ads}$), and desorption/regeneration ($T_{des}$) (Supplementary Table 8). These COPs as well as the working capacities ($\Delta W$) for KMF-1 were first evaluated under the conventional operating conditions of cooling/refrigeration and heating[4] and further compared to those of the benchmark materials for each category of AHP applications. All values for benchmark materials were calculated by applying the same procedure for KMF-1 to literature data[4,11,14,31,40].

Regarding AC applications, COP for cooling ($COP_C$) is defined as the useful energy output ($Q_{evap}$) that is withdrawn by the evaporator, divided by the energy source required as an input ($Q_{des}$) for the regeneration of the adsorbent. Figure 4a reports the calculated $COP_C$ for KFM-1 versus its volumetric working capacity for a given standard refrigeration condition, i.e., $T_{ev} = 5$ °C and $T_{ads} = 30$ °C. At a low driving temperature of 70 °C, KFM-1 with a $COP_C$ of 0.75 outperforms not only SAPO-34 but

also the majority of MOFs, except Co-CUK-1. Typically, when $T_{des}$ exceeds 70 °C, KMF-1 exhibits the highest volumetric working capacities ever attained of 0.358 $mL_{H2O}$ $mL_{MOF}^{-1}$ for AC application (Fig. 4a, c), exceeding the performance of SAPO-34 (0.136 $mL_{H2O}$ $mL_{MOF}^{-1}$) and of the best MOFs reported so far, i.e., Co-CUK-1 (0.346 $mL_{H2O}$ $mL_{MOF}^{-1}$), CAU-10 (0.297 $mL_{H2O}$ $mL_{MOF}^{-1}$), MOF-303 (0.257 $mL_{H2O}$ $mL_{MOF}^{-1}$), MIP-200 (0.165 $mL_{H2O}$ $mL_{MOF}^{-1}$), MIL-160 (0.111 $mL_{H2O}$ $mL_{MOF}^{-1}$), and CAU-23 (0.017 $mL_{H2O}$ $mL_{MOF}^{-1}$). Importantly, KMF-1 keeps a unique attractiveness in terms of working capacity and $COP_C$ for a wide range of cooling conditions, as illustrated in Supplementary Figs. 19 and 20.

Regarding AHP applications, COP for heating ($COP_H$) is defined as the sum of the useful energy output of $Q_{con}$ and $Q_{ads}$ during the adsorption process, divided by the energy input ($Q_{des}$) required for regeneration of the adsorbent. The calculated $COP_H$ and working capacity for KFM-1 are compared with those of benchmark materials under conventional heat pump conditions at $T_{ev} = 15$ °C, $T_{con} = 30$ °C, and $T_{ads} = 45$ °C using a desorption temperature of 85 °C (Fig. 4b). KMF-1 reaches a $COP_H$ (1.74) as high as that of the best water adsorbent so far (Co-CUK-1), however, with a volumetric working capacity (0.353 $mL_{H2O}$ $mL_{MOF}^{-1}$) that largely exceeds the performance of all existing water adsorbents reported so far (Fig. 4d).

Therefore, in light of this careful thermodynamic analysis, KMF-1 appears as a unique multipurpose water adsorbent, maintaining an unprecedented level of performance for both AC and AHP applications. As a further in-depth evaluation of KFM-1 for the target applications, two additional performance indicators

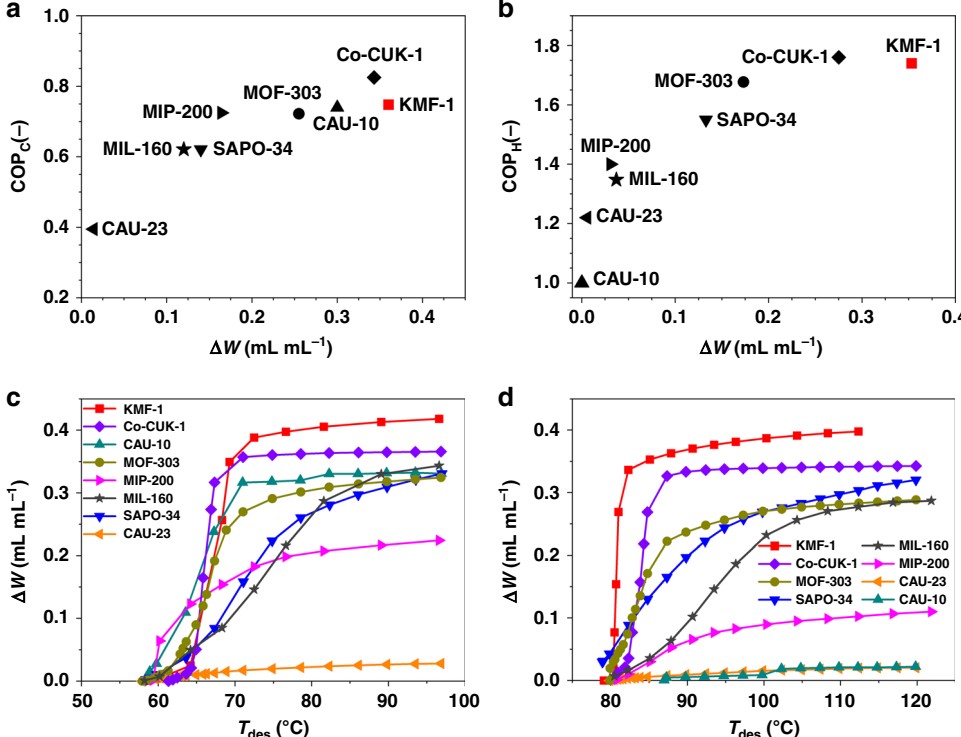

**Fig. 4 Water-sorption performances of KMF-1 and benchmark adsorbents. a** Coefficient of performance for cooling ($COP_C$) versus volumetric working capacity ($\Delta W$) defined as the volume of liquid water per volume of dry adsorbent; these thermodynamic calculations were performed for the standard AC conditions ($T_{ev} = 5$ °C, $T_{ads} = 30$ °C, $T_{cond} = 30$ °C, and $T_{des} = 70$ °C). **b** Coefficient of performance for heating ($COP_H$) versus volumetric working capacity ($\Delta W$) defined as the volume of liquid water per volume of dry adsorbent; these thermodynamics calculations were performed for standard AHP conditions ($T_{ev} = 15$ °C, $T_{ads} = 45$ °C, $T_{cond} = 30$ °C, and $T_{des} = 85$ °C). **c** Plots of volumetric working capacity ($\Delta W$) as a function of desorption temperature ($T_{des}$) for AC conditions ($T_{ev} = 5$ °C, $T_{ads} = 30$ °C, and $T_{cond} = 30$ °C). **d** Plots of volumetric working capacity ($\Delta W$) as a function of desorption temperature ($T_{des}$) for AHP conditions ($T_{ev} = 15$ °C, $T_{ads} = 45$ °C, and $T_{cond} = 30$ °C).

were critically assessed under working conditions: (i) the specific energy capacity that measures the maximum energy transfer capacity of the water adsorbent from the evaporator in cooling conditions[4,8,10], and (ii) the heat-storage capacity which is highly important for daily or seasonal energy-storage application[6,7,18]. These two performance indicators only rarely reported so far for MOFs, were calculated for KMF-1 as well as the best existing water adsorbents for comparison.

Figure 5 and Supplementary Table 6 reveals that under operating conditions at $T_{ev} = 5\text{-}10\,°C$ and $T_{des} = 70\,°C$, KMF-1 is the water adsorbent with the record volumetric (263–266 kWh m$^{-3}$) and gravimetric (244–246 Wh kg$^{-1}$) specific energy capacities. Furthermore, this MOF exhibits exceptional gravimetric (323 Wh kg$^{-1}$) and volumetric (348 kWh m$^{-3}$) heat-storage capacities at $T_{ev} = 10\,°C$, $T_{con} = 30\,°C$, and $T_{des} = 70\,°C$ that are much higher than Co-CUK-1, this level of performances even exceeding that of the conventional energy-storage materials such as zeolite 13X, CaCl$_2$-impregnated mesoporous silica gel, and LiCl-doped multi-wall carbon nanotube, which all show heat-storage capacities below 200 kWh m$^{-3}$ even at higher desorption temperature ($T_{des} = 100\,°C$)[13].

Kinetic sorption behaviors of water adsorbents are decisive for its further promotion as real AHT devices since shorter water-adsorption/desorption cycle time leads to higher power output for cooling and heating. The water-adsorption/desorption profiles of KMF-1 and the best-benchmark MOF adsorbents including Co-CUK-1 and MOF-303 were thus compared in Supplementary Fig. 21. Typically, while KMF-1 and MOF-303 show equivalent adsorption and desorption rates under the same operating conditions, the adsorption rate for KMF-1 is two times faster than for Co-CUK-1 at 30 °C and RH 35%. KMF-1 also outperforms Co-CUK-1 even in the desorption step at 63 °C and RH 10%.

As a leap-forward for the application development, KFM-1 was shaped via wet granulation (Supplementary Method) and deposited on a flat-plate aluminum heat exchanger (Supplementary Fig. 23) to assess its dynamic sorption performances under diverse AC and AHP operating conditions (Supplementary Table 8). Indeed, the specific cooling power (SCP, kW kg$^{-1}$) and specific heating power (SHP, kW kg$^{-1}$), defining the achievable heat transfer rate (d$Q$/d$t$) per gravimetric amounts of adsorbent during one cycle of AC and AHP device, respectively, are key performance metrics to position a

water adsorbent for real applications[46,47]. KMF-1 is highly efficient in adsorption cooling devices operated even at low evaporator temperature ($T_{ev} = 5\,°C$), high condenser temperature ($T_{con} = 40\,°C$), and low regeneration temperatures ($T_{des} = 65\text{-}80\,°C$) (Supplementary Table 10 and Fig. 26). These observations state that KMF-1 can be used at low temperature cooling down to 5 °C as well as at severe climate conditions up to 40 °C. Moreover, KMF-1 is shown to deliver a SCP$_{max}$ up to 3.74 kW kg$^{-1}$ under operating conditions at $T_{ev} = 15\,°C$, $T_{con} = 30\,°C$, and $T_{des} = 70\,°C$ (cycle 3). For comparison, using a similar adsorber configuration, SAPO-34 was reported to show comparable SCP at higher regeneration temperature ($T_{des} = 90\,°C$)[47], however, associated with a lower SCP of about 3 kW kg$^{-1}$ under operating conditions $T_{con} = 20\,°C$ and $T_{ev} = 20\,°C$. Interestingly, kinetics evaluation of KMF-1 under various heating conditions (Supplementary Table 11) revealed its high efficiency for use in AHT devices operated at high condenser temperatures ($T_{con} = 35\text{-}45\,°C$) and relatively low regeneration temperatures ($T_{des} = 75\text{-}85\,°C$) with typically high SHP$_{max}$ values of 3.11–5.34 kW kg$^{-1}$. KFM-1 is as far as we know the first MOF effective for heating applications at high condensation temperature of up to 45 °C.

## Discussion

To the best of our knowledge, KMF-1 was demonstrated to be an ideal multipurpose water adsorbent that outperforms all existing microporous materials including Co-CUK-1 for both adsorption-driven cooling and heating applications. This adsorbent encompasses the optimal water-sorption properties with the highest volumetric working capacities and specific energy capacities ever attained under AC and AHP working conditions combined with very high COP values, excellent cycling stability, and importantly the use of low driving temperatures that paves the way toward an efficient exploitation of solar heat. Moreover, KFM-1 exhibits remarkable heat-storage capacities in a broad range of working conditions. Interestingly, a rationale analysis of the structural/textural/hydrophilic features of a series of promising MOF water adsorbents (Supplementary Table 12) revealed that KMF-1 encompasses the optimal features for AHT applications in terms of pore size ~6 Å, pore volume ~0.5 cm$^3$ g$^{-1}$, moderate adsorption enthalpy ~45-55 kJ mol$^{-1}$, steep water adsorption at $P/P°\sim0.1$ as well as a 1D-channel architecture to favor efficient one-step adsorption and desorption processes. This unprecedented level of performance is coupled with a green synthesis, easy shaping, and scale-up, as well as proven chemical robustness in harsh conditions. The combination of all these features provides a unique opportunity to promote this material at the industrial level to efficiently store thermal energy and achieve both heating and cooling on demand.

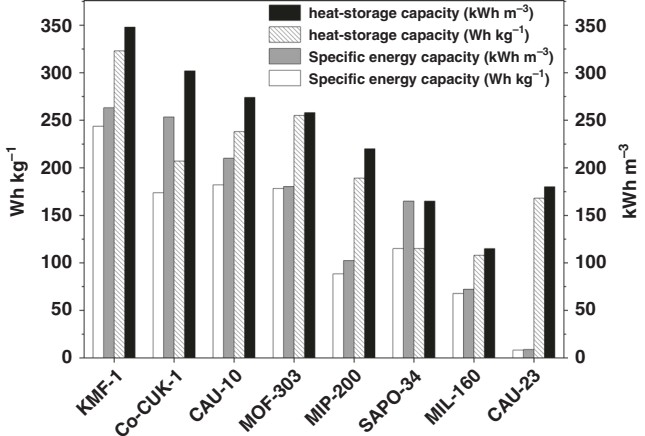

**Fig. 5 Specific energy capacities and heat-storage capacities for KMF-1 and benchmark adsorbents expressed in gravimetry and volumetric scales.** Boundary conditions: heats transferred from the evaporator in one cooling cycle at $T_{ev} = 5\,°C$, $T_{ads} = 30\,°C$, $T_{con} = 30\,°C$, and $T_{des} = 70\,°C$ (white and gray bars) and heat-storage capacity at $T_{ev} = 10\,°C$, $T_{ads} = 30\,°C$, $T_{con} = 30\,°C$, and $T_{des} = 70\,°C$ (dashed and dark gray bars). White and dashed bars denote the gravimetric heats (Wh kg$^{-1}$), while gray and dark gray bars indicate the volumetric heats (kWh m$^{-3}$).

## Methods

**Synthesis of KMF-1.** The aluminum precursor in solution was first prepared by dissolving 3.333 g of Al$_2$(SO$_4$)$_3$·18H$_2$O into 27 mL of deionized water. This solution was then slowly added to the mixture solution constituted by 1.551 g of 2,5-pyrroledicarboxylic and 0.8 g of NaOH in 27 mL of deionized water. After stirring at room temperature for 30 min, the reaction solution was heated and kept at 120 °C for 12 h under reflux condition. The white precipitate of KMF-1 was separated by using filtration and purified with deionized water and ethanol.

**Crystal structure determination.** Synchrotron X-ray diffraction data were collected for the hydrated and anhydrous KMF-1 samples r on beamlines 9B and 2D at the Pohang Accelerator Laboratory (Pohang, Korea) using highly collimated monochromatic synchrotron radiation, i.e., $\lambda = 1.5225$ Å and $\lambda = 0.9000$ Å, respectively. The detector arm of the vertical scan diffractometer consisted of seven sets of Soller slits, flat Ge(111) crystal analyzers, anti-scatter baffles, and scintillation detectors, with each set separated by 20°. Data for hydrated KMF-1 were obtained on the sample at room temperature in flat-plate mode, with a step size of 0.01° for a scan time of 3 s per step, and overlaps of 2° to the next detector bank over the 2$\theta$ range of 5–100°. For anhydrous KMF-1, the sample was obtained in transmission mode as Debye–Sherrer pattern with 66 mm sample-to-detector distance and 30 s exposure time on a Rayonix MX225HS CCD area detector with a

fixed wavelength ($\lambda = 0.90000$ Å) on BL2D-SMC. The PAL BL2D-SMDC program[48] was used for data collection. The Fit2D program[49] was used for the conversion of integrated two-dimensional (2D) patterns to one-dimensional (1D) patterns, wavelength and detector refinement, and the calibration measurement of a National Institute of Standard and Technology (NIST) Si-640c standard sample. Powder samples were packed in a 0.4-mm diameter (wall thickness = 0.01 mm) capillary (Hampton Research Inc. glass number 50). The capillary tube was attached to a custom-made goniometer head with a custom-built vacuum manifold and heated up to 150 ℃ (using Cryojet 5) under vacuum condition. Powder X-ray diffraction patterns were indexed using the DICVOL06 program implemented in the FullProf program suite. Profile refinement of the structure model was performed using the Rietveld method in the JANA2006 package and the GSAS program package. During Rietveld refinement, a pseudo-Voigt function and microscopic broadening, together with a manually interpolated background, were used to describe the peak shapes. KMF-1 was considered as a rigid body, and its thermal motion was described using TLS method[43].

**Measurement of water isotherm and cyclic water adsorption/desorption.**
Water-sorption isotherms were measured by an intelligent gravimetric analyzer (IGA, Hiden Analytical Ltd.). The IGA was automatically operated to precisely control the water vapor pressure (1–95% RH) and temperature (10–100 ℃). Prior to adsorption experiments, samples were fully dehydrated at 150 ℃ for 12 h under high vacuum (<$10^{-6}$ Torr). Multiple cycles of water adsorption–desorption were performed using a thermogravimetric analyzer (TGA, DT Q600, TA Instruments, Universal V4.5 A) connected with a humidity generator. The humidity was controlled by using two mass flow controllers, and a humidified nitrogen gas flow was passed through a thermogravimetric balance. The adsorption profiles were measured at 30 ℃ in humid nitrogen with 35% RH, while desorption data were collected at 70 ℃ and 100 ℃ in dry nitrogen with 4.8% RH, and 1.4% RH, respectively.

**Assessment of thermodynamic performance in AHT application.** The thermodynamic calculations of AC and AHT cycles are performed by an express method based on the methodology reported by De Lange et al.[4]. The COP is adopted to illustrate the energy efficiency of the heat pump cycle from a thermodynamic perspective. The energy analysis allows the determination of the COP, which is a ratio of useful heating or cooling energy output provided to the work required. To be able to assess the COP of a working pair, knowledge of the enthalpy of adsorption is of prime importance[4]. Isosteric heats of adsorption were estimated by the Clausius–Clapeyron equation and compensated by the virial equation[45].

**Kinetic measurements.** The gravimetric Large Temperature Jump (G-LTJ) experimental (Supplementary Figs. 1b, 22, and 23) approach[47] was applied to obtain the kinetic curves (Supplementary Fig. S25) at different boundary conditions for AC and AHT applications using shaped KMF-1 granules (Supplementary Fig. 22 and Table 7). The characteristic sorption time ($\tau$) as well as $SCP_{max}$, $SCP_{80\%}$, $SHP_{max}$, and $SHP_{80\%}$ were calculated on the basis of both kinetic and isobaric sorption measurement data (Supplementary Fig. 25) as explained in more detail in the Supplementary Information section 18.

**Molecular simulations.** The initial structural model of the Al-MOF incorporating PyDC was built in silico, started with the CAU-10 framework. The resulting model was then fully geometry optimized by density functional theory (DFT), including the relaxation of both unit cell and atomic positions. These DFT calculations were carried out within the Gaussian plane waves method as implemented in CP2K package within generalized gradient approximation (GGA) with the Perdew–Burke–Ernzerhof (PBE) exchange functional[50]. The energy cutoff for the plane waves is set at 500 Ry. Goedecker, Teter, and Hutter (GTH) pseudo-potentials[51] and double-zeta-basis sets (DZVP)[52] were employed, while DFT-D3 van der Waals dispersion corrections were considered with a cutoff radius of 10 Å[53]. The density-derived electrostatic and chemical (DDEC) net atomic charges[54] were further computed using the DFT optimized electron density as an input (Supplementary Table 2). Geometric properties of the MOF framework, including the free pore volume, $N_2$-accessible surface area, were computed using the Zeo + + software package[55]. Configurational-bias GCMC simulations were performed to predict the water-adsorption isotherms at $T = 30$ ℃. The simulation box was made of 12 conventional unit cells ($3 \times 2 \times 2$), maintaining the atoms fixed in their initial positions. The interactions between the guest water molecules and the MOF structure were described by a combination of site-to-site LJ contributions and Coulombic terms. The Lennard–Jones (LJ) parameters for all atoms of the MOF framework were taken from the generic UFF forcefield[56]. Following the treatment adopted previously for MIL-160[15], the hydrogen atoms of both the μ-OH moieties and the N–H functions of the organic linkers interact with water via only a Coulombic term. The TIP4P/2005 model[57] was used as a microscopic model to represent the water molecule. Short-term dispersion forces were truncated at a cutoff radius of 12 Å, while the cross-term LJ parameters were calculated by means of the Lorentz–Berthelot combination rule. The long-range electrostatic interactions were handled using the Ewald summation technique. For each point of the adsorption isotherm, $5 \times 10^8$ MC cycles were considered to ensure the convergence. In order to gain insight into the configurational distributions of the water in the MOF, additional data were calculated at different pressure, including the

hydrogen-bond networks and the radial distribution functions (RDF) between the intermolecular atomic pairs of the guests and the MOF framework. Further, the adsorption enthalpy at low coverage was calculated using the standard energy/particle fluctuations in the grand canonical ensemble and the Widom's test particle method[58].

## Data availability
The X-ray crystallographic data for KMF-1 have been deposited at the Cambridge Crystallographic Data Centre (CCDC), under deposition number CCDC 1984701 (KMF-1 anhydrous) and CCDC 1984702 (KMF-1 hydrated). These data can be obtained free of charge from the CCDC via www.ccdc.cam.ac.uk. Further data that support the plots within this paper and other findings of this study are available from the corresponding author upon reasonable request.

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

## Acknowledgements
Korean authors are grateful to the Global Frontier Center for Hybrid Interface Materials of Korea (GFHIM) (Grant No. NRF-2013M3A6B1078879) and the National Research Council of Science & Technology (NST) of Korea (the R&D Convergence Program, CRC-14-1-KRICT) for financial support. X-ray crystallography experiments with synchrotron radiation were performed at the Pohang Accelerator Laboratory. We thank the Center for Computational Engineering and Sciences (FAPESP/CEPID Grant # 2013/08293-7) and CINES (cmc2017 project) for the computational resources.

## Author contributions
K.H.C. and U.-H.L. contributed to the synthesis, shaping and general characterization of KMF-1, and contributed to the writing of the paper. D.D.B. performed the computationally assisted structure determination of KMF-1, the simulation of the water-adsorption isotherms, and the analysis of the mechanism in play, and she equally contributed to the writing of the paper. J.S.L. collected water-sorption data. D.M. collected the synchrotron diffraction data on the powder sample. S.J.C. contributed to the structure determination of KMF-1. J.P. calculated the thermodynamic efficiency of KMF-1 for water sorption. W.L. and A.S. collected kinetic sorption data, analyzed water-sorption kinetics, and evaluate specific cooling and heating power values. G.M. designed and supervised the modeling part of this study and led the writing of the paper. J.-S.C. designed and supervised the study on water sorption, analyzed the data, and led the writing of the paper.

## Competing interests
The authors declare no competing interests.
