## [Peer Review File · Nature Communications]

REVIEWER COMMENTS

Reviewer #1 (Remarks to the Author):

In this study, a new aluminum metal-organic framework (KMF-1) was synthesized and examined its applicability to adsorption-driven cooling/chiller (AC) and adsorption-driven heat pump (AHP) systems. It is an interesting study; however the manuscript is not well described, and thus it is difficult to understand the findings accurately. Also, the performance of KMF-1 does not appear to be as large as claimed in this paper against the currently most promising Co-CUK-1. Therefore, I am afraid that I cannot recommend publication in Nature Communications under the current state. The details are listed below.

1. Since it is not clear how the "rational design" of KMF-1, which has a larger water adsorption capacity than benchmark adsorbents, was conducted, it is desirable that the strategy be clearly explained.
2. Due to the lack of explanation of AC and AHP systems and definition of temperatures T_{ev} , T_{con} , T_{ads} , and T_{des} in each system, it should be difficult for non-specialist readers to understand the performance of KMF-1 described in the manuscript. For example, in the evaluation of specific energy capacity and heat storage capacity of KMF-1 (Fig. 5), only T_{ev} , T_{con} , and T_{des} are provided as the parameters, but it should be hard for many readers to understand why T_{ads} is not shown.
3. The COP(C) of KMF-1 is smaller than that of Co-CUK-1, and the water adsorption capacity of KMF-1 is only about 3% larger than Co-CUK-1. In addition, under COP(H) conditions, the water adsorption capacity of KMF-1 is 1.3 times that of Co-CUK-1, but the COP(H) value of KMF-1 is slightly smaller than that of Co-CUK-1. Moreover, the heat storage capacities per volume of KMF-1 are comparable with those of Co-CUK-1. Therefore, the performance of KMF-1 does not appear to be as great as claimed in this paper. It is thus hoped that more discussion will be given in this paper to reveal the superior performance of KMF-1.
4. As far as the snapshot obtained by GCMC simulation is concerned (Fig. 3), it is considered that the adsorption process is caused by the capillary condensation from the metastable state. In other words, KMF-1 is expected to show a hysteresis in the adsorption-desorption process. However, this study does not show experimental data on adsorption-desorption hysteresis, and thus it is not possible to judge whether the assumed AC or AHP cycle is realistic or not.
5. It is unclear whether the water adsorption properties of the benchmark adsorbents shown in this paper are literature values or were experimentally obtained values under the same conditions as for KMF-1.

Reviewer #2 (Remarks to the Author):

This is a nice piece of work by the Maurin and Chang research groups, in which they discovered a water-stable MOF that have promising water adsorption properties and tested its performance in adsorption chillers (AC) and heat pumps (AHP). The high working capacity with water, good hydrothermal stability and easiness in synthesis of this MOF are all impressive--as the authors discussed in the manuscript, designing a good MOF for water adsorption is always difficult. In addition to the lab-scale adsorption experiments (measuring the adsorption isotherms), they also did thermal & kinetics analysis with AC/AHP prototypes using this MOF as the adsorbent material. They performed very detailed calculations on the cooling efficiency (COP) and heat transfer. These analysis are extremely useful for the assessment and design of larger-scale, practical AC/AHP instruments with this

adsorbent, perhaps eventually leading towards commercialization. Therefore, I recommend this manuscript to be accepted in Nature Communications after minor revisions. My detailed comments are listed below.

1. Hysteresis in the adsorption isotherms is known as a common problem in water adsorption, which can hurt the performance of adsorbent-water AC/AHP pairs both thermodynamically (increase the uptake at desorption condition, thus decrease the working capacity) and kinetically (requires longer cycle times). In this manuscript, the authors provided some relevant information on this (for example, kinetics measurements and recyclability analysis), but the desorption branches in the isotherms were not shown directly. It would be beneficial if those data could be added.

2. The authors compared the performance of KMF-1 with several other MOFs for water adsorption performance, but it's not exactly clear why KMF-1 is the best performer. It would be really nice if the authors can add some discussion on this, which can further increase the value of this paper. For example, is the addition of the amino group the sole reason of KMF-1's better performance than its isoreticular analog CAU-10? Did the (presumably) smaller pore size also matter? Also, what are the distinctive structure features, for example, pore size, pore shape and topology, that made KMF-1 better than other top MOFs for this purpose? Those structural features have been found critical in MOF-based AC/AHP applications and have been used in the design of MOFs with even better performances in a number of closely-related papers, such as ChemMat-2016-28-6243, JACS-2017-139-10601, ChemMat-2019-31-2702, ACSApplNanoMat-2019-2-3050.

3. The authors obtained the isosteric heat of adsorption (Q_{st}) by fitting the experimental isotherms to Virial and C-C equations. However, GCMC simulations can also give the Q_{st} directly. This value should be reported alongside the fitted values.

Reviewer #3 (Remarks to the Author):

The authors presented a combined experimental and computational study for the development of a so-called KMF-1 Al-based new adsorbent for adsorption-driven heat transfer devices. The work appears to be conducted with care and in detail, and the manuscript is well written. The KMF-1 adsorbent was characterized to possess very promising water adsorption properties - a desirable step location of ~ 0.2 P/Po and a large water uptake. The authors also demonstrated that, compared to other sorbents, KMF-1 appears to possess the best performance, despite its COP_c is lower than that of Co-CUK-1. Overall, I think this work should be of interest to the broad materials community, and the material synthesized and reported in this work has the potential for practical applications with energy saving. However, before I could recommend publication of this work, a major revision (with further review) is required to address the following concerns:

1. The biggest concern about the work is that it remains unclear about why KMF-1 outperforms other sorbents. The title of the manuscript is "rational design of ...", but I do not really see how others can move forward, based on the finding in this study, to achieve even better sorbents. What makes KMF-1 so special? Is it due to hydrogen network? But if it is, what's the key factor to enable that? Pore sizes? These insights are critically needed.

2. I am concerned about the claim of "the computationally predictive approach". Although this work showed that the simulated and experimental isotherms seem to agree with each other, the authors should comment on how they can ensure that the simulations have been properly equilibrated. Specially, it seems that the isotherm (simulated) shows a "slower" converging shape (to the saturated loading, or in other words, the step is not very steep), and this might be attributed to the sampling? Also, in the SI, it shows that at 0.02 g/g, one of the channels seems to be kind of filled. This suggests that all other channels might also be saturated at that pressure point. In this sense, would the predicted isotherm be saturated at a lower P/Po? Also a related point, how do the experimental desorption isotherms look like?

3. The authors commented on the aspect of "green" several times (e.g., green synthesis route). Can the authors more quantitatively comment on this aspect? The authors also mentioned the STY value. What's the values for other adsorbents that are included in this study for comparison?

4. I find the section in SI for the kinetic study is not easy to follow. Also, can the authors comment on the dynamic behavior of the adsorption and desorption for other two promising MOFs - MOF303 and Co-CUK-1? Their comparison with KMF-1 may be important, which might yield important insights for the future design of MOFs for adsorption-driven applications.

Overall, this is a good study, and I think it could be accepted after the above comments are satisfactorily addressed.

Reviewer 1:

General comment. In this study, a new aluminum metal-organic framework (KFM-1) was synthesized and examined its applicability to adsorption-driven cooling/chiller (AC) and adsorption-driven heat pump (AHP) systems. It is an interesting study; however, the manuscript is not well described, and thus it is difficult to understand the findings accurately. Also, the performance of KFM-1 does not appear to be as large as claimed in this paper against the currently most promising Co-CUK-1. Therefore, I am afraid that I cannot recommend publication in Nature Communications under the current state. The details are listed below.

Reply. We are grateful to the reviewer for his/her interest to our study and the constructive comments that offered us the opportunity (i) to provide more details on the principles of AC/AHP systems, (ii) to emphasize the outstanding performances of KFM-1 vs the best current water adsorbents by adding complementary experimental data (water adsorption/desorption isotherms and adsorption/desorption rates), (iii) to clarify the origin of the experimental data we reported for the other water adsorbents, and importantly (iv) to explain our computational design approach.

Comment 1. Since it is not clear how the "rational design" of KFM-1, which has a larger water adsorption capacity than benchmark adsorbents, was conducted, it is desirable that the strategy be clearly explained.

Reply 1. We thank the reviewer for this constructive comment that allowed us to elaborate more in the main text our computational design strategy. To summarize, we deliberately built *in silico* a novel isostructural analogue of CAU-10 with the objective to achieve a steep water adsorption isotherm at relative pressure intermediate between $P/P^\circ = 0.05$ and $P/P^\circ = 0.17$ exhibited by the furane-dicarboxylate based CAU-10 derivative (MIL-160) and the benzene-dicarboxylate CAU-10 respectively, while maintaining a water saturation capacity as high as for MIL-160. This optimal target range for P/P° was identified from our previous findings on the best MOF water adsorbents so far, i.e. Co-CUK-1 ($P/P^\circ = 0.12$) of similar pore size than the CAU-10 platform. Indeed, we selected in purpose the 2,5-pyrroledicarboxylate (PyDC) linker instead of the pristine benzene-dicarboxylate linker with the main objective, besides achieving a higher water uptake with the use of a five-membered heterocycle, to fine-tune the hydrophilicity of the pore environment by the introduction of polar -NH functional groups capable of hydrogen bonding interactions expected to be of weaker strength than those established by the bridging μ_2 -OH groups of the inorganic node that rendered MIL-160 too

hydrophilic.

The following sentences were added in the manuscript on page 6:

“Computational Design towards Green Synthesis, Structure Determination and Stability Testing

We deliberately built in silico a novel isostructural analogue of CAU-10 with the objective to achieve a steep water adsorption isotherm at relative pressure intermediate between $P/P^\circ = 0.05$ and $P/P^\circ = 0.17$ exhibited by the furane-dicarboxylate based CAU-10 derivative (MIL-160)¹⁵ and the benzene-dicarboxylate CAU-10¹⁵ respectively, while maintaining a water saturation capacity as high as for MIL-160. This optimal target range for P/P° was identified from our previous findings on the best MOF water adsorbents so far, i.e. Co-CUK-1¹¹ ($P/P^\circ = 0.12$) of similar pore size (~ 6 Å) than the CAU-10 platform. Indeed, we selected in purpose the 2,5-pyrroledicarboxylate (PyDC) linker instead of the pristine benzene-dicarboxylate linker with the main objective, besides achieving a higher water uptake with the use of a five-membered heterocycle, to fine-tune the hydrophilicity of the pore environment by the introduction of polar -NH functional groups capable of hydrogen bonding interactions expected to be of weaker strength than those established by the bridging μ_2 -OH groups of the inorganic node that rendered MIL-160 too hydrophilic.”

Comment 2. Due to the lack of explanation of AC and AHP systems and definition of temperatures T_{ev} , T_{con} , T_{ads} , and T_{des} in each system, it should be difficult for non-specialist readers to understand the performance of KFM-1 described in the manuscript. For example, in the evaluation of specific energy capacity and heat storage capacity of KMF-1 (Fig. 5), only T_{ev} , T_{con} , and T_{des} are provided as the parameters, but it should be hard for many readers to understand why T_{ads} is not shown.

Reply 2. We did provide only a few technical details of AC/AHP systems since their principles are detailed in many more specialized articles we referred in the manuscript. Indeed, we thank the reviewer to give us the opportunity to include a brief explanation of these systems and a definition of the main parameters in play. This is now added on page 3 of the manuscript while a schematic illustration on adsorptive cooling/cooling systems is provided in **Fig. S1**. In addition, all boundary conditions including temperatures (T_{ev} , T_{ads} , T_{con} , and T_{des}) were included in the captions of Fig. 4 and Fig. 5. More specifically, it is generally assumed that the adsorption (T_{ads}) and condensation (T_{con}) temperatures are the same since temperatures of the adsorption bed and the condenser in actual systems are commonly controlled

by external cooling water.

The following sentences were added on page 3 of the manuscript:

“AHT system typically operates under a full cycle of water adsorption/desorption (Supplementary Fig. S1a)^{11,14}. During the adsorption step, heat is taken up from the evaporator (Q_{ev}) leading to a decrease in temperature enabling AC use while as adsorption is exothermic, heat (Q_{ads}) is released to cooling water of the adsorbent bed. When the adsorbent becomes water saturated, regeneration is achieved by the input of useful heat (Q_{des}), and the heat is released to cooling water of condenser (Q_{con}). AHP can thus operate with the released heats of adsorption (Q_{ads}) and condensation (Q_{con}). A detailed thermodynamic T-P diagram of such cycles is provided in Supplementary Fig. S1b⁴. Suitable temperature boundaries for evaporation (T_{ev}), adsorption (T_{ads}), condensation (T_{con}) and desorption (T_{des}) need to be identified to use the full capacity of the water adsorbents for either AC or AHT uses.”

The following Figure S1 was added in the ESI.

Figure S1. a, Typical operation of adsorptive heat transformation (AHT) cycle consisting of adsorption stage (chamber 1) and desorption stage (chamber 2)^{1, 2}. The arrow direction shows the heat transfer while adsorbing or desorbing the water. Low temperature heat is depicted in blue, medium temperature heat in green, and high temperature heat in red. Notation: heat of evaporation Q_{ev} , heat of adsorption Q_{ads} , heat of desorption (or regeneration) Q_{des} , and heat of condensation Q_{con} . b, Typical adsorptive cooling cycle plotted in the P-T diagram³. Notation: temperature and pressure of the

evaporator (T_{ev} , P_{ev}) and the condenser (T_{con} , P_{con}) and desorption temperature (T_{des}).

Comment 3. The COP(C) of KMF-1 is smaller than that of Co-CUK-1, and the water adsorption capacity of KMF-1 is only about 3% larger than Co-CUK-1. In addition, under COP(H) conditions, the water adsorption capacity of KMF-1 is 1.3 times that of Co-CUK-1, but the COP(H) value of KMF-1 is slightly smaller than that of Co-CUK-1. Moreover, the heat storage capacities per volume of KMF-1 are comparable with those of Co-CUK-1. Therefore, the performance of KMF-1 does not appear to be as great as claimed in this paper. It is thus hoped that more discussion will be given in this paper to reveal the superior performance of KMF-1.

Reply 3. We thank the reviewer for this comment emphasizing the need to strengthen the argument for the superior performances of KFM-1 in particular with respect to the best water adsorbent so far, i.e. Co-CUK-1. Indeed, there are several indicators to consider for the evaluation of water adsorbents for adsorptive cooling and heating applications: coefficient of performances (COP_C/COP_H), volumetric working capacities, sorption rates for adsorption & desorption and long-term cycle stability. Here below, we provide a comparative systematic evaluation of the performances of KFM-1 and Co-CUK-1.

(i) COPs. We fully agree with the reviewer that **KFM-1 and Co-CUK-1 show similar level of performances.** Indeed, KMF-1 reaches a COP_H (1.74) as high as that of Co-CUK-1 (1.76). Regarding COP_C , as far as we know, a commercial target value is 0.65 at optimized conditions. In this regard, COP_C (0.75) of KMF-1 while slightly lower than the value of Co-CUK-1 (0.82) is sufficiently high compared to the commercial adsorbent, SAPO-34 ($COP_C = 0.64$) at $T_{ev} = 5\text{ }^\circ\text{C}$, $T_{ads} = 30\text{ }^\circ\text{C}$, $T_{cond} = 30\text{ }^\circ\text{C}$, and $T_{des} = 70\text{ }^\circ\text{C}$.

(ii) Working capacities. **KMF-1 outperforms Co-CUK-1.** Indeed the volumetric working capacity of KMF-1 ($0.358\text{ ml}_{H_2O}\text{ ml}_{MOF}^{-1}$) for cooling application is only slightly higher than the performance of Co-CUK-1 ($0.346\text{ ml}_{H_2O}\text{ ml}_{MOF}^{-1}$) at $T_{ev} = 5\text{ }^\circ\text{C}$, $T_{ads} = 30\text{ }^\circ\text{C}$, $T_{cond} = 30\text{ }^\circ\text{C}$, and $T_{des} = 70\text{ }^\circ\text{C}$. Regarding heating application, its volumetric working capacity ($0.353\text{ ml}_{H_2O}\text{ ml}_{MOF}^{-1}$) largely exceeds the performance of all existing water adsorbents including Co-CUK-1 ($0.275\text{ ml}_{H_2O}\text{ ml}_{MOF}^{-1}$) at $T_{ev} = 15\text{ }^\circ\text{C}$, $T_{ads} = 45\text{ }^\circ\text{C}$, $T_{cond} = 30\text{ }^\circ\text{C}$, and $T_{des} = 85\text{ }^\circ\text{C}$.

(iii) Adsorption and desorption rates. **KMF-1 outperforms Co-CUK-1.**

Typically, as shown below (**Fig. R1**), the adsorption rate for KMF-1 is typically 2 times faster than for Co-CUK-1 at 30°C and RH 35%. KMF-1 is also more efficient than Co-CUK-1 even in the desorption step. This additional set of experimental data is now provided in a new figure in the ESI (Fig. S21) with a comparative kinetics study between KMF-1 and two benchmark adsorbents, i.e. Co-CUK-1 and MOF-303.

Fig. R1. (a) Gravimetric and (b) volumetric adsorption-desorption profiles of KMF-1 and Co-CUK-1. These profiles were collected from the second adsorption-desorption cycle as a function of time measured by TGA: adsorption at 30 °C and RH 35 % and desorption at 63 °C and RH 10 % in a nitrogen flow (100 ml/min). The second cycle profiles were obtained just after the first profiles of fully dehydrated adsorbents for adsorption at 30 °C and RH 35 %, followed by desorption at 63 °C and RH 10 %. The sample weight was determined by full dehydration at 150 °C under a dry nitrogen flow (100 ml/min). The ramping rate of desorption temperature is 20 °C/min.

(iv) Long-term cycle stability. Both **KMF-1** and **Co-CUK-1** show exceptional long-term stability during multi-cyclic water adsorption desorption experiments without loss of initial capacities up to 50th cycles. This is illustrated in the figure provided below (**Fig. R2**).

Fig. R2. Thermogravimetric analysis profile for long-term stability test under 50 cycles of water adsorption-desorption of (a) KMF-1 and (b) Co-CUK-1 (taken from our previous paper: Lee JS, *et al. ACS Appl Mater Interfaces* **11**, 25778-25789 (2019)). Test conditions: adsorption at 30 °C in humid nitrogen (35% RH) and desorption at 63 °C in nitrogen with low humidity (6% RH). Prior to the multiple cycle experiment, the first cycle was carried out by a different condition such that Co-CUK-1 is dehydrated at 150 °C for 1 hour in dry N₂, hydrated at 30 °C in humid nitrogen (35% RH), and then dehydrated again at 63 °C in nitrogen with low humidity (6% RH).

(v) Specific energy/Heat storage capacities. **KMF-1 outperforms Co-CUK-1** since this new MOF exhibits the record volumetric (263-266 kWh m⁻³) and gravimetric (244-246 Wh kg⁻¹) specific energy capacities. Furthermore, this MOF exhibits exceptional gravimetric (323 Wh kg⁻¹) and volumetric (348 kWh m⁻³) heat storage capacities as already shown in Fig. 5 and Table S5.

In addition to these standard performance metrics, KFM-1 shows more attractive features than Co-CUK-1 in terms of preparation:

(vi) Synthesis yields and conditions. **KMF-1 is more favourable than Co-CUK-1.** For the aqueous phase synthesis (Table R1), yield (93%) and space-time-yield (68 kg/m³·day) of KMF-1 are much higher than those of Co-CUK-1 (67%, 24 kg/m³·day). Further the synthesis temperature (200°C) of Co-CUK-1 is higher than that (120°C) of KMF-1. The synthetic conditions for all benchmark adsorbents are now summarized in ESI with an additional (Table S1).

Therefore the examination of these six criteria demonstrates the superior performances of KMF-1 vs the best water adsorbent so far (Co-CUK-1) for both AC and AHP applications.

Table R1. Comparison of synthesis conditions and yields of KMF-1 and Co-CUK-1

MOF	molar ratio of precursors	Reaction Temp. (°C)	Reaction time (h)	Yield (%)	Space-time to Yield (kg/m ³ /day)
KMF-1	1 Al ₂ (SO ₄) ₃ : 1 PyDC: 2.5 NaOH	~ 120 (reflux)	12 h	93	68
Co-CUK-1	3 CoCl ₂ : 2 PDC: 3 KOH	200	15 h	67	24

Abbreviation of chemicals: PyDC (2,5-pyrroledicarboxylic acid); PDC (2,4-pyridine dicarboxylic acid).

To address this comment, the following Table (Table S1) and Figure (Figure S21) were added in the ESI and referred in the manuscript accordingly.

Table S1. Comparison of synthesis conditions and yields of selected water adsorbents

MOF	Reactant mole ratio	Reaction Temp. (°C)	Reaction time (h)	Yield (%)	Space-time to Yield (kg/m ³ /day)	Reference
KMF-1	1 Al ₂ (SO ₄) ₃ : 1 PyDC: 2.5 NaOH	~ 120 (reflux)	12 h	93	68	This work
Co-CUK-1	3 CoCl ₂ : 2 PDC: 3 KOH	200	15 h	67	24	1
MIP-200	3.74 ZrCl ₄ : 1.25 H4mdip: 663FA: 370 AAn	120	48-72 h	96	7.2	2
CAU-10	1.5 Al ₂ (SO ₄) ₃ : 1 NaAlO ₂ : 4 IPA: 8 NaOH	Reflux	15 h	95	93	4
CAU-23	3 AlCl ₃ : 1 NaAlO ₂ : 4 TDC: 8 NaOH	Reflux	4 h	84	137	5
MIL-160	1Al(OH)(CH ₃ COO) ₂ : 1 FDCA	Reflux	24 h	93	185	6
MOF-303	1AlCl ₃ : 1PyrzDC : 1.5 NaOH	~100 (reflux)	24 h	35	4	7
MOF-801P	1ZrOC ₂ : 1 FUA: 37 FA: 52 DMF	120	12 h	63	148	8

Abbreviation of chemicals: PyDC (2,5-pyrroledicarboxylic acid); IPA (isophthalic acid); TDC (2,5-thiophenedicarboxylic acid); FDCA (2,5-furandicarboxylic acid); PyrzDC (3,5-pyrazoledicarboxylic acid); FUM (fumaric acid); FA (formic acid); DMF (N,N-dimethylformamide); AAn (Acetic anhydride); PDC (2,4-pyridine dicarboxylic acid); H4mdip

(3,3',5,5'-tetracarboxydiphenylmethane).

Fig. S21. (a) Gravimetric and (b) volumetric adsorption-desorption profiles of KMF-1, Co-CUK-1, and MOF-303. These profiles were plotted from the second adsorption-desorption cycle as a function of time, measured by TGA: adsorption at 30 °C and RH 35 % and desorption at 63 °C and RH 10 % in a nitrogen flow (100 ml/min). The second cycle profiles were obtained just after the first profiles of fully dehydrated adsorbents for adsorption at 30 °C and RH 35 %, followed by desorption at 63 °C and RH 10 %. The sample weight was determined by full dehydration at 150 °C under a dry nitrogen flow (100 ml/min). The ramping rate of desorption temperature is 20 °C/min.

The following sentences were added in the manuscript :

on page 5 : *"Furthermore, its synthesis conditions and product yields are still far to be optimal (Supplementary Table S1)."*

on page 8 : *"A relatively good STY over 68 kg m⁻³ day⁻¹ was obtained after simple purification with water and ethanol which is much higher than the value previously reported for the best water adsorbent Co-CUK-1 so far (24 kg m⁻³ day⁻¹) (Supplementary Table S1)."*

on page 17: *"Kinetic sorption behaviors of water adsorbents are decisive for its further promotion as real AHT devices since shorter water adsorption/desorption cycle time leads to higher power output for cooling and heating. The water adsorption/desorption profiles of KMF-1 and the best benchmark MOF adsorbents including Co-CUK-1 and MOF-303 were thus compared in Fig. S21. Typically, while KMF-1 and MOF-303 show equivalent adsorption and desorption rates under the same operating conditions, the*

adsorption rate for KMF-1 is 2 times faster than for Co-CUK-1 at 30°C and RH 35%. KMF-1 also outperforms Co-CUK-1 even in the desorption step at 63°C and RH 10 %.

on page 18: "KMF-1 was demonstrated to be ideal multi-purpose water adsorbent that outperforms all existing microporous materials including Co-CUK-1 for both adsorption-driven cooling and heating applications."

Comment 4. As far as the snapshot obtained by GCMC simulation is concerned (Fig. 3), it is considered that the adsorption process is caused by the capillary condensation from the metastable state. In other words, KMF-1 is expected to show a hysteresis in the adsorption-desorption process. However, this study does not show experimental data on adsorption-desorption hysteresis, and thus it is not possible to judge whether the assumed AC or AHP cycle is realistic or not.

Reply 4. We thank the reviewer for allowing us to clarify the adsorption/desorption behaviour of KFM-1. To address this comment, the desorption isotherm at 40 °C was added in Fig. 2a. This additional data clearly demonstrates that the adsorption-desorption isotherms do not show any hysteresis. The fully reversible water adsorption-desorption behaviour of KMF-1 is also observed in the multi-cyclic water adsorption/desorption experiments we provided as Fig. S17 in the initial ESI. Indeed, very similar levels of working capacities were obtained which is ascribed to the full reversibility of water adsorption-desorption cycles in dynamic conditions. These thermodynamics and dynamics data clearly support the reliability of the AC/AHP cycle discussed in the manuscript.

The following sentence and Figure 2a were modified on page 10 *"The water adsorption behaviour of KMF-1 was first explored at three different temperatures (20-40 °C) (Fig. 2a) revealing a S-shaped adsorption isotherm with a full reversible desorption branch."*

Fig. 2 | Water sorption isotherms for KMF-1 and benchmark adsorbents. **a**, Experimental water adsorption isotherms for KMF-1 collected at three different temperatures: 20 °C (■), 30 °C (●), and 40 °C (▲). Insert shows the adsorption/desorption isotherms at 40 °C. **b**, Comparison of water adsorption isotherms for KMF-1 (■) and benchmark adsorbents: MOF-303 (●)⁴⁰, CAU-10 (▲)⁴, SAPO-34 (▼)⁴, MIP-200 (►)¹⁴, Co-CUK-1 (◆)¹¹ at 30 °C and CAU-23 (◄)³¹ at 25 °C.

Comment 5. It is unclear whether the water adsorption properties of the benchmark adsorbents shown in this paper are literature values or were experimentally obtained values under the same conditions as for KMF-1.

Reply 5 To address this comment, more details on how the water adsorption properties (page 10) and thermodynamic evaluations (page 13) of benchmark adsorbents were obtained are now incorporated in the manuscript.

On page 10: “all these corresponding data being extracted from the literature“

On page 13: “All values for benchmark materials were calculated by applying the same procedure for KMF-1 to literature data^{4,11,14,31,40}.”

Reviewer 2:

General comment. This is a nice piece of work by the Maurin and Chang research groups, in which they discovered a water-stable MOF that have promising water adsorption properties and tested its performance in adsorption chillers (AC) and heat pumps (AHP). The high working capacity with water, good hydrothermal stability and easiness in synthesis of this MOF are all impressive-as the authors discussed in the manuscript, designing a good MOF for water adsorption is always difficult. In addition to the lab-scale adsorption experiments (measuring the adsorption isotherms), they also did thermal & kinetics analysis with AC/AHP prototypes using this MOF as the adsorbent material. They performed very detailed calculations on the cooling efficiency (COP) and heat transfer. These analysis are extremely useful for the assessment and design of larger-scale, practical AC/AHP instruments with this adsorbent, perhaps eventually leading towards commercialization. Therefore, I recommend this manuscript to be accepted in Nature Communications after minor revisions. My detailed comments are listed below.

Reply. We are grateful to the reviewer for his/her very positive assessment and for the very minor comments that allowed us to strengthen the importance of our contribution by expanding our design strategy as well as by adding complementary experimental data (water adsorption/desorption data) as simulated water adsorption enthalpy.

Comment 1. Hysteresis in the adsorption isotherms is known as a common problem in water adsorption, which can hurt the performance of adsorbent-water AC/AHP pairs both thermodynamically (increase the uptake at desorption condition, thus decrease the working capacity) and kinetically (requires longer cycle times). In this manuscript, the authors provided some relevant information on this (for example, kinetics measurements and recyclability analysis), but the desorption branches in the isotherms were not shown directly. It would be beneficial if those data could be added.

Reply 1. We thank the reviewer for allowing us to clarify the adsorption/desorption behaviour of KFM-1. To address this comment, the desorption isotherm at 40 °C was added in Fig. 2a. This additional data clearly demonstrates that the adsorption-desorption isotherms do not show any hysteresis. The fully reversible water adsorption-desorption behaviour of

KMF-1 is also observed in the multi-cyclic water adsorption and desorption experiments we provided as Fig. S17 in the initial ESI. Indeed, very similar levels of working capacities were obtained which is ascribed to the full reversibility of water adsorption-desorption cycles in dynamic conditions. These thermodynamics and dynamics data clearly support the reliability of the AC/AHP cycle discussed in the manuscript.

The following sentence and Figure 2a were modified on page 10 “*The water adsorption behaviour of KMF-1 was first explored at three different temperatures (20-40 °C) (Fig. 2a) revealing a S-shaped adsorption isotherm with a full reversible desorption branch.*”

Fig. 2 | Water sorption isotherms for KMF-1 and benchmark adsorbents. **a**, Experimental water adsorption isotherms for KMF-1 collected at three different temperatures: 20 °C (■), 30 °C (●), and 40 °C (▲). Insert shows the adsorption/desorption isotherms at 40 °C. **b**, Comparison of water adsorption isotherms for KMF-1 (■) and benchmark adsorbents: MOF-303 (●)⁴⁰, CAU-10 (▲)⁴, SAPO-34 (▼)⁴, MIP-200 (►)¹⁴, Co-CUK-1 (◆)¹¹ at 30 °C and CAU-23 (◄)³¹ at 25 °C.

Comment 2. The authors compared the performance of KMF-1 with several other MOFs for water adsorption performance, but it's not exactly clear why KMF-1 is the best performer. It would be really nice if the authors can add some discussion on this, which can further increase the value of this paper. For example, is the addition of the amino group the sole reason of KMF-1's better performance than its isorecticular analog CAU-10? Did the (presumably)

smaller pore size also matter? Also, what are the distinctive structure features, for example, pore size, pore shape and topology, that made KMF-1 better than other top MOFs for this purpose? Those structural features have been found critical in MOF-based AC/AHP applications and have been used in the design of MOFs with even better performances in a number of closely-related papers, such as ChemMat-2016-28-6243, JACS-2017-139-10601, ChemMat-2019-31-2702, ACSAppiNanoMat-2019-2-3050.

Reply 2. We thank the reviewer for this constructive comment that allowed us to elaborate more in the main text our computational design strategy. To summarize, we deliberately built *in silico* a novel isostructural analogue of CAU-10 with the objective to achieve a steep water adsorption isotherm at relative pressure intermediate between $P/P^\circ = 0.05$ and $P/P^\circ = 0.17$ exhibited by the furane-dicarboxylate based CAU-10 derivative (MIL-160) and the benzene-dicarboxylate CAU-10 respectively, while maintaining a water saturation capacity as high as for MIL-160. This optimal target range for P/P° was identified from our previous findings on the best MOF water adsorbents so far, i.e. Co-CUK-1 ($P/P^\circ = 0.12$) of similar pore size than the CAU-10 platform. Indeed, we selected in purpose the 2,5-pyrroledicarboxylate (PyDC) linker instead of the pristine benzene-dicarboxylate linker with the main objective, besides achieving a higher water uptake with the use of a five-membered heterocycle, to fine-tune the hydrophilicity of the pore environment by the introduction of polar -NH functional groups capable of hydrogen bonding interactions expected to be of weaker strength than those established by the bridging μ_2 -OH groups of the inorganic node that rendered MIL-160 too hydrophilic.

We further made a rationale analysis of the structural/textural/hydrophilic features of a series of promising MOF water adsorbents (see new Table incorporated as Table S12 in the ESI). Interestingly, this revealed that KMF-1 encompasses the optimal features for AHT applications in terms of pore size ~ 6 Å, pore volume ~ 0.5 cm³/g, moderate adsorption enthalpy ~ 45 -55 kJ.mol⁻¹, steep water adsorption at $P/P^\circ \sim 0.1$ as well as a 1D-channel architecture to favour efficient 1-step adsorption and desorption processes.

To address this comment the following sentence was added in the conclusion section of the manuscript (on page 18):

"Interestingly, a rationale analysis of the structural/textural/hydrophilic features of a series of promising MOF water adsorbents (see Table S12) revealed that KMF-1 encompasses the optimal features for AHT applications in terms of pore size ~ 6 Å, pore volume ~ 0.5 cm³/g, moderate adsorption enthalpy ~ 45 -

55 kJ.mol⁻¹, steep water adsorption at P/P°~0.1 as well as a 1D-channel architecture to favour efficient 1-step adsorption and desorption processes”

Table S12 related to the Comparison of the textural/structural/hydrophilic features of the best water adsorbent MOFs along with their performances for AHT applications, was added in the ESI.

Comment 3. The authors obtained the isosteric heat of adsorption (Qst) by fitting the experimental isotherms to Virial and C-C equations. However, GCMC simulations can also give the Qst directly. This value should be reported alongside the fitted values.

Reply 3. We added the GCMC simulation data on page 13 of the manuscript with the following sentence:

«This value is consistent with the adsorption enthalpy predicted by GCMC simulations at 0.1 g g⁻¹ (-54 kJ.mol⁻¹) suggesting a facile regeneration of KMF-1 ».

The following sentence was also added on page 22 of the manuscript:

“Further, the adsorption enthalpy was calculated using the Widom insertion method. “

Reviewer 3:

General comment. The authors presented a combined experimental and computational study for the development of a so-called KMF-1 AI-based new adsorbent for adsorption-driven heat transfer devices. The work appears to conduct with care and in detail, and the manuscript is well written. The KMF-1 adsorbent was characterized to possess very promising water adsorption properties - a desirable step location of ~ 0.2 P/P₀ and a large water uptake. The authors also demonstrated that, compared to other sorbents, KMF-1 appears to possess the best performance, despite its COP_c is lower than that of Co-CUK-1. Overall, I think this work should be of interest to the broad materials community, and the material synthesized and reported in this work has the potential for practical applications with energy saving. Overall, this is a good study, and I think it could be accepted after the above comments are satisfactorily addressed.

Reply. We are grateful to the reviewer for his/her interest on our study and the very positive assessment of the paper as well as for the constructive comments that offer us the opportunity to clarify the computational strategy and deliver more details on the green synthesis as well as complementary thermodynamics and kinetics adsorption data.

Comment 1. The biggest concern about the work is that it remains unclear about why KMF-1 outperforms other sorbents. The title of the manuscript is "rational design of ...", but I do not really see how others can move forward, based on the finding in this study, to achieve even better sorbents. What makes KMF-1 so special? Is it due to hydrogen network? But if it is, what's the key factor to enable that? Pore sizes? These insights are critically needed.

Reply 1. We thank the reviewer for this constructive comment that allowed us to elaborate more in the main text our computational design strategy. To summarize, we deliberately built *in silico* a novel isostructural analogue of CAU-10 with the objective to achieve a steep water adsorption isotherm at relative pressure intermediate between $P/P^{\circ} = 0.05$ and $P/P^{\circ} = 0.17$ exhibited by the furane-dicarboxylate based CAU-10 derivative (MIL-160) and the benzene-dicarboxylate CAU-10 respectively, while maintaining a water saturation capacity as high as for MIL-160. This optimal target range for P/P° was identified from our previous findings on the best MOF water adsorbents so far, i.e. Co-CUK-1 ($P/P^{\circ} = 0.12$) of similar pore size than the CAU-10 platform. Indeed, we selected in purpose the 2,5-pyrroledicarboxylate (PyDC) linker instead of the pristine benzene-dicarboxylate linker with the main

objective, besides achieving a higher water uptake with the use of a five-membered heterocycle, to fine-tune the hydrophilicity of the pore environment by the introduction of polar -NH functional groups capable of hydrogen bonding interactions expected to be of weaker strength than those established by the bridging μ_2 -OH groups of the inorganic node that rendered MIL-160 too hydrophilic.

We further made a rationale analysis of the structural/textural/hydrophilic features of a series of promising MOF water adsorbents (see new Table incorporated as Table S12 in the ESI). Interestingly, this revealed that KMF-1 encompasses the optimal features for AHT applications in terms of pore size ~ 6 Å, pore volume ~ 0.5 cm³/g, moderate adsorption enthalpy ~ 45 -55 kJ.mol⁻¹, steep water adsorption at $P/P^\circ \sim 0.1$ as well as a 1D-channel architecture to favour efficient 1-step adsorption and desorption processes.

To address this comment the following sentence was added in the conclusion section of the manuscript (on page 18):

"Interestingly, a rationale analysis of the structural/textural/hydrophilic features of a series of promising MOF water adsorbents (see Table S12) revealed that KMF-1 encompasses the optimal features for AHT applications in terms of pore size ~ 6 Å, pore volume ~ 0.5 cm³/g, moderate adsorption enthalpy ~ 45 -55 kJ.mol⁻¹, steep water adsorption at $P/P^\circ \sim 0.1$ as well as a 1D-channel architecture to favour efficient 1-step adsorption and desorption processes"

Table S12 related to the Comparison of the textural/structural/hydrophilic features of the best water adsorbent MOFs along with their performances for AHT applications, was added in the ESI.

Comment 2. I am concerned about the claim of "the computationally predictive approach". Although this work showed that the simulated and experimental isotherms seem to agree with each other, the authors should comment on how they can ensure that the simulations have been properly equilibrated. Specially, it seems that the isotherm (simulated) shows a "slower" converging shape (to the saturated loading, or in other words, the step is not very steep), and this might be attributed to the sampling? Also, in the SI, it shows that at 0.02 g/g, one of the channels seems to be kind of filled. This suggest that all other channels might also be saturated at that pressure point. In this sense, would the predicted isotherm be saturated at a lower P/P_o ?

Also a related point, how do the experimental desorption isotherms look like?

Reply 2. We thank the reviewer for this suggestion. To carefully check the convergence for our simulated water adsorption isotherm, Configuration-Bias GCMC scheme with $5 \cdot 10^8$ MC steps was considered to revisit the water adsorption isotherm. The resulting water adsorption isotherm is reported in Figure S3 and does not show anymore any convergence issues and confirm that the steep increase of water amount occurs at $P/P^0 = 0.12$. To show the simulation is now properly equilibrated, we added the error bars on the simulated adsorption isotherm.

Figure S3 is provided in ESI:

Figure S3. Comparison between the GCMC predicted (black) and experimental (red) water adsorption isotherms for KFM-1 at 30 °C. Error bars are included for the simulations.

The following sentence was added on the molecular simulation section on page 22.

“Configurational-bias GCMC simulations were performed to predict the water adsorption isotherms at T=30 °C.”

In addition, the desorption isotherm at 40 °C was added in Fig. 2a. This

additional data clearly demonstrates that the adsorption-desorption isotherms do not show any hysteresis. The fully reversible water adsorption-desorption behaviour of KMF-1 is also observed in the multi-cyclic water adsorption and desorption experiments we provided as Fig. S17 in the initial ESI. Indeed, very similar levels of working capacities were obtained which is ascribed to the full reversibility of water adsorption-desorption cycles in dynamic conditions. These thermodynamics and dynamics data clearly support the reliability of the AC/AHP cycle discussed in the manuscript.

Fig. 2 | Water sorption isotherms for KMF-1 and benchmark adsorbents. **a**, Experimental water adsorption isotherms for KMF-1 collected at three different temperatures: 20 °C (■), 30 °C (●), and 40 °C (▲). Inset contains the adsorption and desorption isotherms at 40 °C. **b**, Comparison of water adsorption isotherms for KMF-1 (■) and benchmark adsorbents: MOF-303 (●)⁴⁰, CAU-10 (▲)⁴, SAPO-34 (▼)⁴, MIP-200 (►)¹⁴, Co-CUK-1 (◆)¹¹ at 30 °C and CAU-23 (◄)³¹ at 25 °C.

Comment 3. The authors commented on the aspect of "green" several times (e.g., green synthesis route). Can the authors more quantitatively comment on this aspect? The authors also mentioned the STY value. What's the values for other adsorbents that are included in this study for comparison?

Reply 3 To address carefully this comment, a new table is provided in ESI (**Table S1**) which compares the synthesis conditions and synthesis yields/STY of KFM-1 with other benchmark adsorbents. In terms of synthesis yields and synthetic conditions, KMF-1 is demonstrated to be more favourable than Co-

CUK-1 and MOF-303. For the aqueous phase synthesis, synthesis yield (93%) and space-time-yield ($68 \text{ kg/m}^3\text{day}$) of KMF-1 are much higher than those of Co-CUK-1 (67%, $24 \text{ kg/m}^3\text{day}$) and MOF-303 (30%, $4 \text{ kg/m}^3\text{day}$). Furthermore, the synthesis temperature (200°C) of Co-CUK-1 is much higher than that (120°C) of KMF-1.

The following Table (**Table S1**) was added in the ESI:

Table S1. Comparison of synthesis conditions and yields of selected water adsorbents

MOF	Reactant mole ratio	Reaction Temp. ($^\circ\text{C}$)	Reaction time (h)	Yield (%)	Space-time to Yield ($\text{kg/m}^3\text{/day}$)	Reference
KMF-1	1 $\text{Al}_2(\text{SO}_4)_3$: 1 PyDC: 2.5 NaOH	~ 120 (reflux)	12 h	93	68	This work
Co-CUK-1	3 CoCl_2 : 2 PDC: 3 KOH	200	15 h	67	24	1
MIP-200	3.74 ZrCl_4 : 1.25 H4mdip: 663FA: 370 AAn	120	48-72 h	96	7.2	2
CAU-10	1.5 $\text{Al}_2(\text{SO}_4)_3$: 1 NaAlO_2 : 4 IPA: 8 NaOH	Reflux	15 h	95	93	4
CAU-23	3 AlCl_3 : 1 NaAlO_2 : 4 TDC: 8 NaOH	Reflux	4 h	84	137	5
MIL-160	1 $\text{Al}(\text{OH})(\text{CH}_3\text{COO})_2$: 1 FDCA	Reflux	24 h	93	185	6
MOF-303	1 AlCl_3 : 1 PyrDC : 1.5 NaOH	~ 100 (reflux)	24 h	35	4	7
MOF-801P	1 ZrOC_2 : 1 FUA: 37 FA: 52 DMF	120	12 h	63	148	8

Abbreviation of chemicals: PyDC (2,5-pyrroledicarboxylic acid); IPA (isophthalic acid); TDC (2,5-thiophenedicarboxylic acid); FDCA (2,5-furandicarboxylic acid); PyrDC (3,5-pyrazoledicarboxylic acid); FUM (fumaric acid); FA (formic acid); DMF (N,N-dimethylformamide); AAn (Acetic anhydride); PDC (2,4-pyridine dicarboxylic acid); H4mdip (3,3',5,5'-tetracarboxydiphenylmethane).

The following sentence was completed on page 7 of the manuscript:

« A relatively good STY over $68 \text{ kg m}^{-3} \text{ day}^{-1}$ was obtained after simple purification with water and ethanol which is much higher than that the value previously reported for the best water adsorbent Co-CUK-1 ($24 \text{ kg m}^{-3} \text{ day}^{-1}$) (Supplementary Table S1). »

Comment 4. I find the section in SI for the kinetic study is not easy to follow. Also, can the authors comment on the dynamic behavior of the adsorption and desorption for other two promising MOFs - MOF303 and Co-CUK-1? Their comparison with KMF-1 may be important, which might yield important insights for the future design of MOFs for adsorption-driven applications.

Reply 4. We thank the reviewer for this comment that allowed us to clarify the ESI and add complementary data on the adsorption kinetics. We added a comparative kinetics study between KMF-1 and two benchmark adsorbents, i.e. Co-CUK-1 and MOF-303 in terms of gravimetric and volumetric sorption profiles as a function of time provided as a new figure (**Fig. S21**). It comes that KMF-1 outperforms Co-CUK-1 in terms of adsorption rate. When we compared the adsorption rates between KMF-1 and Co-CUK-1 at 30 °C and RH 35 %, we found that the adsorption rate for KMF-1 is 2 times faster than that for Co-CUK-1. KMF-1 is also more efficient than Co-CUK-1 even in the desorption stage. KMF-1 and MOF-303 show similar in terms of adsorption and desorption rates.

Fig. S21. (a) Gravimetric and (b) volumetric adsorption-desorption profiles of KMF-1, Co-CUK-1, and MOF-303. These profiles were plotted from the second adsorption-desorption cycle as a function of time, measured by TGA: adsorption at 30 °C and RH 35 % and desorption at 63 °C and RH 10 % in a nitrogen flow (100 ml/min). The second cycle profiles were obtained just after the first profiles of fully dehydrated adsorbents for adsorption at 30 °C and RH 35 %, followed by desorption at 63 °C and RH 10 %. The sample weight was determined by full dehydration at 150 °C under a dry nitrogen flow (100 ml/min). The ramping rate of desorption temperature is 20 °C/min.

The following paragraph was added on page 17 of the manuscript

“Kinetic sorption behaviors of water adsorbents are decisive for its further promotion as real AHT devices since shorter water adsorption/desorption cycle time leads to higher power output for cooling and heating. The water adsorption/desorption profiles of KMF-1 and the best benchmark MOF adsorbents including Co-CUK-1 and MOF-303 were thus compared in Fig. S21. Typically, while KMF-1 and MOF-303 show equivalent adsorption and desorption rates under the same operating conditions, the adsorption rate for KMF-1 is 2 times faster than for Co-CUK-1 at 30°C and RH 35%. KMF-1 also outperforms Co-CUK-1 even in the desorption step at 63°C and RH 10 %.”

REVIEWERS' COMMENTS:

Reviewer #1 (Remarks to the Author):

The manuscript has been improved and all the issues I raised were appropriately addressed, and thus I think that the paper can be published. However, there is a minor comment on the calculation of adsorption enthalpy of water using the Widom insertion method, which is, as far as I know, a method of calculating free energy. It is hoped that a brief explanation about the calculation of enthalpy from the obtained free energy will be provided in the "Molecular simulations" section.

Reviewer #2 (Remarks to the Author):

My comments have been well addressed and I recommend this manuscript to be published.

Reviewer #3 (Remarks to the Author):

The authors have satisfactorily addressed all my previous comments, and therefore I would now recommend publication of this study.

Reviewer 1:

Comment. The manuscript has been improved and all the issues I raised were appropriately addressed, and thus I think that the paper can be published. However, there is a minor comment on the calculation of adsorption enthalpy of water using the Widom insertion method, which is, as far as I know, a method of calculating free energy. It is hoped that a brief explanation about the calculation of enthalpy from the obtained free energy will be provided in the "Molecular simulations" section.

Reply. We thank the reviewer for his/her comment. For clarification, the adsorption enthalpy was calculated from both the standard energy/particles fluctuations in the grand canonical ensemble and the Widom's test particle method which implies the use of test (guest) molecule inserted at random position in the MOF and an average in the NVT ensemble over all positions of the guest. The following text was modified in the manuscript as follows on page 22 with the addition of 1 reference (reference 58):

« Further, the adsorption enthalpy at low coverage was calculated using the standard energy/particles fluctuations in the grand canonical ensemble and the Widom's test particle method⁵⁸»